# Impact Resistance of a Fiber Metal Laminate Skin Bio-Inspired Composite Sandwich Panel with a Rubber and Foam Dual Core

**DOI:** 10.3390/ma16010453

**Published:** 2023-01-03

**Authors:** Wenping Zhang, Ruonan Li, Quanzhan Yang, Ying Fu, Xiangqing Kong

**Affiliations:** 1School of Civil Engineering, Liaoning University of Technology, Jinzhou 121001, China; 2School of Material, Shenyang Ligong University, Shenyang 110000, China; 3Songshan Lake Material Laboratory, Dongguan 523808, China

**Keywords:** FML, bio-inspired composite sandwich panel, low-velocity impact, dual-core, impact resistance, finite element analysis

## Abstract

This paper reports the development of a novel bio-inspired composite sandwich panel (BCSP) with fiber metal laminate (FML) face sheets and a dual core to improve the low-velocity impact behavior based on the woodpecker’s head layout as a design template. The dynamic response of BCSP under impact load is simulated and analyzed by ABAQUS/Explicit software and compared with that of the composite sandwich panel (CSP) with a single foam core. The impact behavior of BCSP affected by these parameters, i.e., a different face sheet thickness, rubber core thickness and foam core height, was also reported. The results show that BCSP has superior impact resistance compared to CSP, with a lower damage area and smaller deformation, while carrying a higher impact load. Concurrently, BCSP is not highly restricted to any particular region when dealing with stress distributions. Compared to CSP, the bottom skin maximum stress value of BCSP is significantly reduced by 2.4–6.3 times at all considered impact energy levels. It is also found that the impact efficiency index of BCSP is 4.86 times higher than that of CSP under the same impact energy, indicating that the former can resist the impact load more effectively than the latter in terms of overall performance. Furthermore, the impact resistance of the BCSP improved with the increase in face sheet thickness and rubber core thickness. Additionally, the height of the foam core has a notable effect on the energy absorption, while it does not play a significant role in impact load. From an economic viewpoint, the height of the foam core retrofitted with 20 mm is reasonable. The results acquired from the current investigation can provide certain theoretical reference to the use of the bio-inspired composite sandwich panel in the engineering protection field.

## 1. Introduction

Composite sandwich structures have been widely used in aerospace, automotive, construction and other fields due to their outstanding performance in heat dissipation, light weight, noise reduction, impact resistance, energy absorption, and so on [1,2]. However, sandwich structures are generally rather sensitive to the damage of impact [3], because the event of impact is instantaneous and the corresponding load amplitude can be many times its static equivalent load [4], which results in a significant reduction in the strength and stiffness of the structure, both externally and internally. For this reason, new strategies and designs for impact protection to improve impact resistance of sandwich structures need to be actively proposed and refined.

Composite sandwich structures consist of a middle lightweight core component that is sandwiched between the top and bottom face sheets. The top and bottom face sheets of composite sandwich structures act as a protective layer against external loads. There are many common face sheet types for composite sandwich panels [5,6,7,8,9,10], e.g., wood materials, metal materials, fiber reinforced plastics composite materials, etc. In recent years, a new face sheet, fiber metal laminate (FML), which combines the performance advantages of metal alloy and composite materials, has attracted much attention due to its excellent impact resistance and high specific strength/stiffness [11,12,13,14]. Many researchers have recently focused on the mechanical properties of composite sandwich structures with FML face sheets [15,16,17,18,19,20,21,22]. Ding et al. [23] proposed experimental and theoretical research on the failure modes of composite sandwich beams with FML face sheets under bending loads. They reported that the composite sandwich beams with FML face sheets have better ultimate load bearing capacity than composite sandwich beams with fiber reinforced plastic (FRP) face sheets. Similarly, Liu et al. [24,25] conducted a drop hammer impact experiment to investigate failure modes and impact resistance characteristics of a composite sandwich panel with FML face sheets. Their results showed that the impact resistance of the composite sandwich panel with FML face sheets was superior to that of a composite sandwich panel with FRP face sheets. Ahmadi et al. [26] used 3D finite element software ABAQUS/Explicit to numerically analyze the impact response of the composite sandwich structure with FML face sheets and found that FML face sheets can increase the energy absorption capability of the sandwich structures.

The above studies show that some progress has been made in the dynamic response of composite sandwich structures with FML face sheets subjected to impact loading. However, the research mentioned above mainly focuses on the sandwich structures with a single core, which still has some defects under impact load, such as stress concentration and shear failure, forming the weakest link in the design [27,28,29,30]. Recently, researchers have also been inspired by nature, applying biometrics to traditional single core sandwich structures to improve their impact resistance [31,32,33]. Typically, Sabah et al. [34] prepared a new type of bio-inspired honeycomb sandwich beam combined with carbon fiber reinforced plastic (CFRP) skins incorporated with a rubber and honeycomb dual core. These bio-inspired sandwich beams with a dual core showed superior impact resistances to that of sandwich beams with a single honeycomb core. Compared to sandwich beams with a single honeycomb core, the peak load of bio-inspired sandwich beams with a dual core is significantly increased by 126–200% at all considered impact energy levels. Such advantage had also been showcased in the repeated impact loading environment [35], with a 125% increase in the specific energy absorption [36]. To further reveal the impact response of dual-core sandwich structures, especially the deformation model and energy absorption capacity, more investigation should be carried out.

From the aforementioned literature, it is distinctly known that various core configurations, skin types and many beneficial bio-inspired designs can further improve the impact resistance of sandwich structures. In order to combine the advantages of FMLs with the advantages of the bio-inspired dual core sandwich structure, this paper proposes a new type of bio-inspired composite sandwich panel (BCSP) comprising FML face sheets incorporated with a rubber and foam dual core based on the woodpecker’s head layout as a design template. Among them, FML face sheets act as a protective layer against external loads. The rubber core spreads and regulates the impact excitation. Simultaneously, the foam core can further absorb higher impact energy. Therefore, the ductility, carrying capacity and impact resistance of the bio-inspired dual-core sandwich panel with FML skins are highly modified. To reveal the impact response of the novel BCSP, finite element models of the BCSP under different impact energies were established by using the ABAQUS 2017/explicit nonlinear analysis software. The failure morphologies, impact load, stress transmission characteristics and energy absorption mechanism of BCSP were observed and compared with those of the composite sandwich panel (CSP) with a single foam core through numerical simulation. The effects of structural parameters, such as the thickness of rubber core, the skin’s thickness and the height of the foam core, on the impact resistance of the BCSP were studied.

## 2. Design of the BCSP

### 2.1. BCSP Development

Woodpeckers, the doctors of trees, often look for hidden bugs in trunks. When they are working, they slam their hard beaks against the tree trunk up to 20 times per second, without causing any damage to their brain. Previous studies have found that woodpeckers have a remarkable ability to absorb shocks [37]. It is well known that in real-world materials science, there is a general tradeoff between hardness and toughness. However, woodpeckers’ skull structures are both tough and extremely resilient, reducing the impact of external transmission to the brain [38]. In principle, the head of a woodpecker can be structurally divided into four major parts: skull bone, sponge bone, hyoid bone and beak, as illustrated in Figure 1a [39]. This study was inspired by the structural characteristics of the woodpecker head, and a new impact resistance layout was designed to improve the existing composite sandwich structure, as illustrated in Figure 1b. The beak of the woodpecker is represented with a FML as the external protective layer of the BCSP to resist load. The hyoid bone is represented by a rubber layer as the first layer of the core for the BCSP, to spread and adjust impact load. The structure of the spongy bone is similar to the Al foam layer as a second core layer for further absorbing higher impact energy. Finally, the BCSP is completed with another FML skin representing the woodpecker’s skull bone.

### 2.2. Geometry Description

The geometric details of the BCSP lay-up are given next. The FML face sheets in the BCSP lay-up itself (both outer skins) consisted of Al 5005 (Table 1) and E glass/epoxy (Table 2). The lay-up scheme of FML face sheets was 1/1. This meant that one outer Al 5005 layers and one E glass/epoxy inlayers in a (0/90) stacking sequence. The FML layer had a total thickness of 1.4 mm. The rubber core employed a thickness of 3 mm, and the Al foam core adopted a thickness of 20 mm. The properties of the material are summarized in Table 3. Generally, the dimension of BCSP was 150 mm × 150 mm. The size of the BCSP was the same as that of CSP, which is based on the experiments reported by Liu [25].

## 3. Numerical Simulation

### 3.1. FE Modeling and Boundary Conditions

In order to gain an in-depth understanding of the dynamic response of the BCSP under impact loading, a high-accuracy FE model of the BCSP was constructed using the non-linear FE software ABAQUS/Explicit, and compared with CSP, as depicted in Figure 2. The face sheet, foam core, and rubber core were meshed using the 8-node hexahedral linear reduction integral element of type C3D8R. The choice of these elements is due to the reduction of computational effort, provided that a fine mesh is adopted. This choice usually does not reduce the accuracy of the analysis, provided that an instability mesh control is simultaneously applied. Additionally, this element type has been shown to effectively simulate the layers of the sandwich structure [1,4,34,35]. Considering the plastic deformation of largeness in the central region where the impactor was impacted, an element size of 1 mm was set after a mesh sensitivity analysis (Figure 3), while coarse meshes were applied at the edge of the BCSP.

The hemispherical impactor’s motion was dominated by a reference point defined in the rigid body, and the mass (8.048 kg) and initial velocity were distributed to the reference point of the impactor. It is fully fixed except for the vertical direction. Furthermore, the peripheries of the BCSP model were stipulated with the setting such that rotations *θ*= 0 (rotations about the *x*-, *y*- and *z*-axis, respectively) and translations, *u*_y_ = *u*_z_ = 0 (translations in the *y*-and *z*-axis, respectively). 

The tie constraint is used to connect the layers of the BCSP to each other. Additionally, a general contact prescription was established to model the interaction between the top of the FML skin of the impactor with the upper surface. The coefficient of Coulomb friction in the tangential direction is taken as 0.3, with hard contact in the normal direction.

### 3.2. Material Properties and Modeling

#### 3.2.1. Material Models of the FML Face Sheets

The FML face sheets consisted of a metal layer (Al 5005) and a composite layer (E glass/epoxy). Firstly, the ductile criterion was chosen in this paper to describe the fracture behavior of the face sheets of the metal layer Al 5005. This ductile damage model has been proved to be a valid method to model the fracture responses of ductile metallic materials during the penetration process. Damage initiates when the damage state variable $\omega_{D}$ is reached: (1)$$\omega_{D}=\int \frac{d{\bar{\varepsilon }}^{pl}}{{\bar{\varepsilon }}_{D}^{pl}(\eta ,{\dot{\bar{\varepsilon }}}^{pl})}=1$$
where $\eta ,\;{\dot{\bar{\varepsilon }}}^{pl},\;{\bar{\varepsilon }}^{pl},\;{\bar{\varepsilon }}_{D}^{pl}$ denote the stress triaxiality, strain rate, equivalent plastic strain and the ultimate value of equivalent plastic strain, respectively. Once damage evolution begins at any point in the material, the stiffness of the material needs to decrease exponentially. The removal of elements indicates that the stiffness degradation has reached a critical value.

Then, in order to predict the damage behavior of the E glass/epoxy composite laminate of FML face sheets, 3D Hashin and Puck failure criteria, which have been used as computational models for predicting damage in composites due to their ease of use and conceptual simplicity [42,43], were employed in this study. The four failure modes are shown in Equations (2)–(5).

Hashin fiber tensile failure mode $\left(\sigma_{11}\geq 0\right)$:(2)$${\left(\frac{\sigma_{11}}{X_{1t}}\right)}^{2}+{\left(\frac{\sigma_{11}}{S_{12}}\right)}^{2}+{\left(\frac{\sigma_{11}}{S_{13}}\right)}^{2}=1$$

Hashin fiber compressive failure mode $\left(\sigma_{11}<0\right)$:(3)$${\left(\frac{\sigma_{11}}{X_{1c}}\right)}^{2}=1$$

Puck matrix tensile failure mode $\left(\sigma_{22}+\sigma_{33}\geq 0\right)$:(4)$${\left(\frac{\sigma_{11}}{2X_{1t}}\right)}^{2}+\frac{\sigma_{22}{}^{2}}{\left|X_{2t}X_{2c}\right|}+{\left(\frac{\sigma_{12}}{S_{12}}\right)}^{2}+\sigma_{22}(\frac{1}{X_{2t}}+\frac{1}{X_{2c}})=1$$

Puck matrix compressive failure mode $\left(\sigma_{22}+\sigma_{33}<0\right)$:(5)$${\left(\frac{\sigma_{11}}{2X_{1t}}\right)}^{2}+\frac{\sigma_{22}{}^{2}}{\left|X_{2t}X_{2c}\right|}+{\left(\frac{\sigma_{12}}{S_{12}}\right)}^{2}+\sigma_{22}(\frac{1}{X_{2t}}+\frac{1}{X_{2c}})=1$$
where $\sigma_{11},\sigma_{22}$ and $\sigma_{12}$ are the stress components of the material integral point, respectively; $X_{1t}{,}^{}X_{2t}$ and $X_{3t}$ are the tensile strength of the E glass/epoxy in directions 1,2 and 3, respectively; $X_{1c}{,}^{}X_{2c}$ and $X_{3c}$ are the E glass/epoxy composite’s compressive strength in directions $1{,}^{}2$ and 3, respectively; $S_{12}{,}^{}S_{13}$ and $S_{23}$ are the E glass/epoxy composite’s shear strength in planes $12{,}^{}13$ and 23, respectively. The material parameters for the E glass/epoxy materials used to calculate the stiffness matrix in the simulation are presented in Table 2.

The macro-mechanical constitutive model of the E glass/epoxy was established by Fortran language based on a user-defined material subroutine interface of ABAQU/Explicit, as demonstrated in Figure 4. When the material meets any of the failure criteria, damage of the material begins to occur, and the stiffness of the material begins to degrade. In the subprogram of this paper, the damage evolution of materials is carried out in the form of mutation. When the damage variable is 1, the element fails completely and is deleted, and the stiffness of the material degrades to 0. In the end, the above state variables are resubmitted to the ABAQUS solver for subesquent calculations.

#### 3.2.2. Material Models of the Core

A crushable foam model, also known as the Deshpande–Fleck model [44], was used for the aluminum foam core, which reduces the foam metallic material to an isotropic reinforced intrinsic model. The *CRUSHABLE FOAM and CRUSHABLE FOAM HARDENING options in the ABAQUS package were used to describe the plastic crushable behavior of Al foam cores. 

The rubber core is defined as a hyperelastic material, using an isotropic option, which is a special case of the Cauchy material description. The rubber material was expressed based on the Mooney–Rivlin model provided as
(6)$$U=C_{10}(\bar{1}-3)+C_{01}(\bar{2}-3)+\frac{1}{D_{1}}{\left(J-1\right)}^{2}$$
where $C_{10}{,}^{}C_{01}{,}^{}D_{1},$ and J are the material parameters and volume ratio, respectively. In addition, 1, 2 and 3 in $\bar{1}=J^{-2/3}1{,}^{}\bar{2}=J^{-4/3}2$ are the strain invariants. In this study, the function of the rubber core is to uniformly distribute the stresses induced by the impact event in order to protect the subsequent layers of the BCSP with a minimum stress, similar to the woodpecker’s head mechanism. When the damage criteria are met, the element deletion function in ABAQUS/Explicit is performed on the rubber core.

### 3.3. Validation of the CSP Numerical Model

As there is no experimental study on the dynamic response of this new type of bio-inspired foam sandwich panel in the available literature, this numerical approach has been validated by a series of experiments on the response of CSP under impact loading [25]. The finite element simulation of the CSP was verified using ABAQUS/Explicit with the same finite element modeling, material properties, boundary conditions, and user-defined subroutines as in Section 3.1 and Section 3.2. Figure 5 shows the comparison of CSP’s simulated and experimental impact energy absorption with different foam core thickness. The results illustrate that at the same initial velocity, the difference between the numerical model predicted values of energy absorption and the test result is the smallest when the thickness of the Al foam is 20mm. The energy absorption of the CSP obtained by experimental result is 133.7 J [25] and the energy absorption of our numerical model subject is 139.3 J. The difference between them is only 4.2%. When the thickness of the Al foam is 35 mm, the energy absorption error between the experimental measured value and numerical model predicted values is the largest, which is 7.8%. Although there are some errors, the above errors are within the allowable range.

Figure 6 shows the comparison of failure modes between the CSP test and simulation under 72 J impact energy. It was found that the Al 5005 layer underwent bending and cracking, and the E-glass layer underwent breakage in the top FML face sheets. The Al foam core underwent compression deformation. This is in favorable agreement with the test results. Concurrently, it is clear to see that the predicted damaged area was 20.81 mm, and this was very close to the measured value of 19.75 mm from the test; the difference between the two is 5%. The predicted deformation depth was 17.86 mm, and this also agreed well with the experiment depth of 18.87 mm; the difference between the two is 5.3%. In summary, numerical models can effectively predict the damage deformation of CSP [25].

Overall, the numerical model displays good agreement with experimental measurements in terms of damage patterns, deformation and energy absorption, thus demonstrating the capability of the modeling approach. The present numerical model can be effectively used to investigate the impact behavior of BCSP.

## 4. Results and Discussion

The above finite element model is used to explore the dynamic response of the BCSP under various impact energies. Additionally, the predicted impact damage, stress transmission, force history and energy absorption mechanism are assessed, and compared with that of the CSP. 

### 4.1. Failure Modes

The failure morphologies of BCSP with different impact energies were demonstrated in Figure 7. It is obvious to see that at all energy levels, the structural damage of the BCSP is mainly concentrated in the contact collision area between the impactor and the BCSP. During the impact process, the E glass/epoxy composite layer matrix cracked, and the Al 5005 metal layer collapsed permanently at the impact position in the top FML face sheets. The deformation of the Al foam core during collision is mainly manifested as local buckling deformation near the upper FML face sheets, and there was no peeling between FML skins and the core layer. At the same time, when the energy is small, the top FML face sheets of BCSP is less damaged, and the Al foam core only deforms slightly in the impact area. As the impact energy increases, the damage area of the top FML face sheets increases obviously and the compression degree of the Al foam core increases. Consequently, the degree of densification of the Al foam core layer further increases, the load-bearing capacity of the BCSP increases, and more impact energy is consumed. 

In addition, compared with the CSP (shown in Figure 5), it can be seen that the BCSP showed less damage and lower deformation under the same impact energy. The CSP experienced localized compression deformation of the Al foam core in the impact area that was severe. Nevertheless, the BCSP experiences less deformation due to the addition of rubber and, thus, its stiffness increased. Therefore, it is apparent that the BCSP provides superior impact damage resistance compared to the existing CSP.

Figure 8 displays the comparison of the damage area between the BCSP and the CSP under different impact energies. The damage area of the BCSP was decreased by 30–60% due to the addition of a layer of rubber core in all impact levels compared with the CSP. This is because the rubber core can transfer stress away from the impact area, thus avoiding stress concentration. Therefore, it can be clearly observed that the BCSP has superior impact resistance compared to the CSP.

### 4.2. Stress Transmission

The propagation paths of stress waves in the BCSP are, in turn, the top FML face sheets, the rubber core, the Al foam core and the bottom FML surface material. Therefore, in terms of safety, the less stress on the underlying face sheets, the less chance it has to reach failure and, thus, is better able to maintain its structural function. Figure 9 depicts the maximum stresses at the bottom face sheets of the CSP and BCSP under different impact energies. As the impact energy increases, it can be clearly observed that the bottom stress values in both panels increase. Furthermore, the maximum stresses in the impact area of the bottom face sheets of the CSP and BCSP are compared, as listed in Table 4. Compared to the CSP, the bottom skin maximum stress value of the BCSP is significantly reduced by 2.4–6.3 times at all considered impact energy levels. The reason for this is the inclusion of a rubber core, which improves the bending resistance and stiffness of the BCSP and allows better adjustment of the stresses away from the bottom face sheets. 

### 4.3. Load History

Impact loading is known to be one of the key factors in quantifying the dynamic response of structures and exploring the failure mechanisms of impact events. Figure 10 shows the predicted impact load history curves of the BCSP for various impact energies. It is evident that the trend of impact load history curves under various impact energies is consistent. Firstly, the impact load increases linearly, and the impact energy at this stage is mainly absorbed by the elastic deformation of the upper face sheets and the compressive deformation of the Al foam core. Secondly, with the increase in impact time, after reaching the ultimate strength of the top face sheets, the E glass/epoxy composite layer matrix crushing damage begins to occur and the Al 5005 metal layer collapses permanently at the impact position in the top FML face sheets. The Al foam core also produces slight buckling, which results in slight fluctuations in the impact force. Finally, as the impact load continues, the intermediate Al foam core continues to bear the impact load. After reaching the peak force, the impact energy is continuously consumed, the punch begins to rebound, and the impact force gradually decreases to 0.

Furthermore, it can be also seen from Figure 10 that the impact load history curves of the BCSP and CSP show the same trend. Additionally, the peak load of the BCSP is higher than the CSP under the same impact energy (72 J). Table 4 summarizes the peak load of the CSP and BCSP under the same impact energy (72 J). It can be seen that the peak load of the BCSP is 9.52 kN, with an increase by about 21% compared to the CSP. This is due to the inclusion of a rubber core, which significantly increases the bending stiffness of the BCSP and improves the load-bearing capacity of the BCSP. In summary, it can be seen that the load-bearing capacity of the BCSP is superior to that of the CSP.

### 4.4. Absorbed Energy

Concurrently, in order to further understand the dynamic response of the BCSP during the whole impact process, the energy absorption history curve also provides a good indication of the mechanical properties of the BCSP under impact load.

The energy absorptions of both panels obtained from numerical simulations are shown in Figure 11 and Table 4. It can be obviously seen that with the increase in impact energy, the absorption capacity of the BCSP increases. The BCSP energy absorption values of impact energies of 44 J, 72 J and 100 J are 31.2 J, 59.02 J and 89.6 J, with an increase of 11%, 18.7% in the energy absorption rate compared to the impact energy of 44 J, respectively. Concurrently, the energy absorption efficiency of the CSP is about 84.2% under an impact energy of 72 J; it is around 5.2% higher than the BCSP. The reason is that the CSP suffers greater damage and, thus, absorbs more energy. However, this capability is undesirable for the continuous use of the structure, which affects the service life of the structure. Therefore, it can be shown that the CSP is worse than the BCSP in resisting further impact loads. 

In order to ensure a fair and meaningful assessment of the impact resistance of both panels, a dimensionless impact resistance efficiency index related to mass, peak force, energy absorption, and other factors was introduced in this study. The impact resistance efficiency index [34], $I_{e}$, is denoted as
(7)$$I_{e}=\frac{E_{abs}F_{\max}}{tg\sigma_{\max}Am}$$
where $E_{\mathrm{abs}}$ is the energy absorption, $F_{\max}$ is the peak load, t is the thickness of the panel, g refers to the gravitational acceleration, $\sigma_{\max}$ is the bottom face sheets maximum stress in the panel, A is the damage area, and m is the mass of the panel. Since the comparison is now normalized by the thickness and mass of the composite sandwich panel, it is fairer to focus the comparison only on the performance of the sample per unit thickness and per unit mass. Therefore, the use of $I_{e}$ enables a fair comparison between the two composite sandwich panels. Taking the impact energy of 72 J as an example, the CSP’s and BCSP’s $I_{e}$ are calculated as 36.21 and 176.23, respectively, and the $I_{e}$ of the BCSP is about 4.86 times that of the CSP. This shows that under the same conditions, the impact resistance of the CSP is much inferior to that of the BCSP. In summary, it is also proved that the effect of using a double-layer core bio-inspired structure in the composite sandwich panel on improving the damage resistance of the composite sandwich panel is obvious.

## 5. Parameter Analysis

Previous studies have shown that the optimal design of the topological structure, shape and size parameters of sandwich structures play an important role in obtaining the best mechanical properties of traditional sandwich structures in different practical applications and different objectives [45,46]. In order to deeply understand the impact resistance of BCSP, this section studies the effect of face sheet thickness, rubber core’s thickness, and Al foam core’s height on the dynamic response of BCSP under impact load.

### 5.1. Effect of the Face Sheet Thickness

The existing research shows that the thickness of the face sheets has an important effect on the impact resistance of the sandwich structure [4,46]. To investigate the effect of the thickness of the face sheets on the impact resistance of the BCSP, BCSPs with four different face sheet thicknesses (*T_f_* = 0.9, 1.2, 1.5 and 1.8 mm) are simulated under an impact energy of 60 J. The peak load and impact energy absorption with the different face sheet thickness are graphed in Figure 12. As can be seen, the peak load of *T_f_* = 0.9, 1.2, 1.5 and 1.8 mm is 7.89 kN, 8.6 kN, 9.17 kN and 10.1 kN, with an increase of 8.9%, 16.2% and 28% in the peak load compared to *T_f_* = 0.9 mm, respectively, which indicates that the BCSP with higher structural stiffness could provide higher impact resistance. Additionally, the energy absorption of *T_f_* = 0.9, 1.2, 1.5 and 1.8 mm is 48.9 J, 45.6 J, 42.3 J and 38.64 J, with an increase of 26.5%, 18% and 9.7% in the energy absorption compared to *T_f_* = 1.8 mm, respectively. At the same time, as the thickness of the face sheets increases, the energy absorption effect gradually decreases and changes significantly. In summary, it can be seen that as the thickness of the face sheet increases, the stiffness of the BCSP will increase, and the energy absorption effect will decrease. 

### 5.2. Effect of the Rubber Core Thickness

To investigate the influence of the rubber core thickness on the impact resistance of the BCSP, four thicknesses of rubber core (i.e., *t* = 1, 2, 3 and 4 mm) are adopted. Figure 13 plots the peak load and energy absorption of the BCSP with four different thickness of the rubber core subjected to an impact energy of 60 J. It is clear that the peak load increases with increasing thickness of the rubber core. This is because the stiffness of the BCSP increases with the thickness of the rubber core, and its impact resistance is improved. Similarly, the energy absorption increases when the rubber core thickness is increased from 2 mm to 4 mm. However, when the rubber core thickness increases from 1 mm to 2 mm, the energy absorption of the BCSP decreases by 6.8%, from 48.63 J to 45.52 J. This is because the bearing capacity is reduced and the structure is seriously damaged when the thickness of the rubber core is thin. Therefore, the thickness of the rubber core should not be too small. When the rubber core thickness is increased from 2 mm to 3 mm, the energy absorption of the BCSP is increased by 7.4%, from 45.52 J to 48.9 J, while when the rubber core thickness is increased from 3 mm to 4 mm, the energy absorption only increases by 4.9%, from 48.8 J to 51.2 J. Therefore, by considering the energy absorption effectiveness, as well as the cost, for the BCSP in this case, the rubber core thickness of 3 mm is adequate in improving the impact resistance. 

### 5.3. Height of the Al Foam Core

In practical engineering [1], the height of the Al foam used to protect structures is usually 20–50 mm. Therefore, the influence of the Al foam core height (*H* = 20, 30, 40 and 50 mm) on the impact response of the BCSP is studied in this part. In Figure 14, it is clearly demonstrated that the peak load and energy absorption of the BCSP are at the same impact energy level. The results present that the peak load of the BCSP increases with the increase in the Al foam height, but the change in the Al foam height has little effect on the peak load of the BCSP under the impact load. The peak load of *H* = 20, 30, 40 and 50 mm is 8.21 kN, 8.43 kN, 8.49 kN and 8.56 kN, with an increase of 2.6%, 3.4% and 4.2% in the peak load compared to *H* = 20 mm, respectively; the peak load varies within 5%. The energy absorption of the BCSP decreases gradually with the increase in the foam core height. The above study shows that the BCSP with a 20 mm Al foam core height displayed superior impact resistance, consistent with the energy absorption index, absorbing about 81% of the impact energy. Therefore, from the perspective of cost, for the BCSP in this case, the Al foam core height of 20 mm is adequate in improving the impact resistance.

## 6. Conclusions

A novel BCSP with FML and a rubber and Al foam dual core based on the configuration of a woodpecker’s head was established for improving low-velocity impact behavior. The impact response of the proposed BCSP structure at different energy levels was examined numerically using the finite element solver ABAQUS/Explicit and compared with CSP. At the same time, the effects of face sheets, rubber core thickness and foam core height on impact peak load and energy absorption of BCSP were further investigated by using this model. The main conclusions are summarized as follows:(1)The results illustrated that the E glass/epoxy composite layer matrix cracked and the Al 5005 metal layer collapsed permanently at the impact position in the top FML face sheets at all impact energy levels. Additionally, compared with the CSP, it can be seen that the BCSP showed less damage and lower deformation due to their higher stiffness under the same impact energy. The damages area of the BCSP was decreased by 30–60% due to the addition of a rubber core layer in all impact levels compared with the CSP. Therefore, it is apparent that the BCSP provides superior impact damage resistance than the existing CSP.(2)The bottom skin maximum stress value of the BCSP was significantly reduced by 2.4–6.3 times compared with the CSP. Similarly, the peak load of the BCSP was 9.52 kN, with an increase by about 21% compared to the CSP under the same impact energy (72 J). Additionally, was also obviously observed that with the increase in impact energy, the absorption capacity of BCSP increases. Furthermore, it was found that the impact efficiency index of the BCSP is 176.23, which is 4.86 times higher than that of the CSP under the same impact energy, indicating that the former can resist the impact load more effectively than the latter in terms of overall performance.(3)In succession, comprehensive parametric studies on the BCSP were carried out by considering various effect parameters under impact loads. As the face sheets increased, the stiffness of the BCSP increased, its impact resistance could be improved. The thickness of the rubber core also had a significant influence on the impact response of the BCSP. Additionally, the rubber core thickness had a significant effect on the energy absorption, while it did not play a big role in the impact load. By considering the energy absorption effectiveness, as well as cost, for the BCSP in this case, the rubber core thickness of 3 mm is adequate to improve the impact resistance. The height of the Al foam core was also found to have an obvious effect on the dynamic response of the BCSP under the impact loads. As the Al foam core height increases, the peak load of the BCSP changes slightly, but the energy absorption of the BCSP improves significantly. From an economic viewpoint, the height of the foam core retrofitted with 20 mm is reasonable.

## Figures and Tables

**Figure 1 materials-16-00453-f001:**
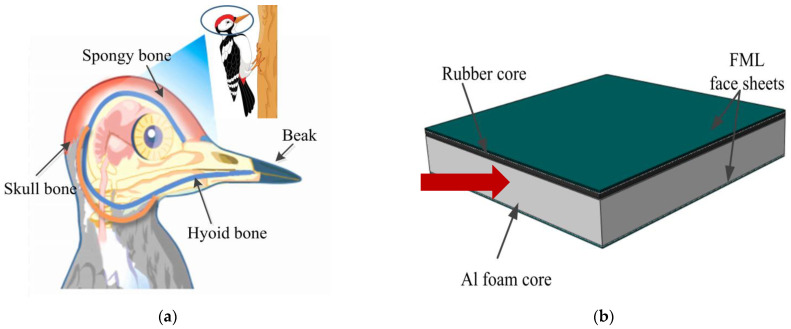
(**a**) Woodpecker head configuration [39]; (**b**) BCSP.

**Figure 2 materials-16-00453-f002:**
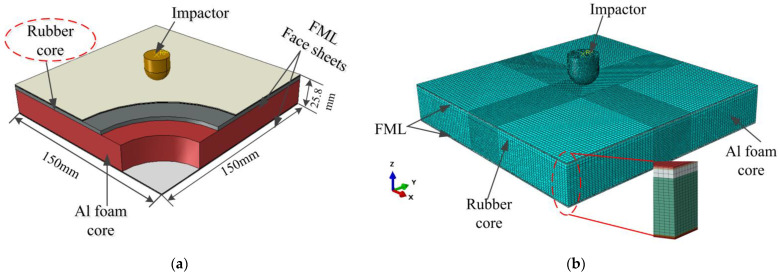
(**a**) Schematic of the geometric configurations of the BCSP. (**b**) Finite element model of the BCSP.

**Figure 3 materials-16-00453-f003:**
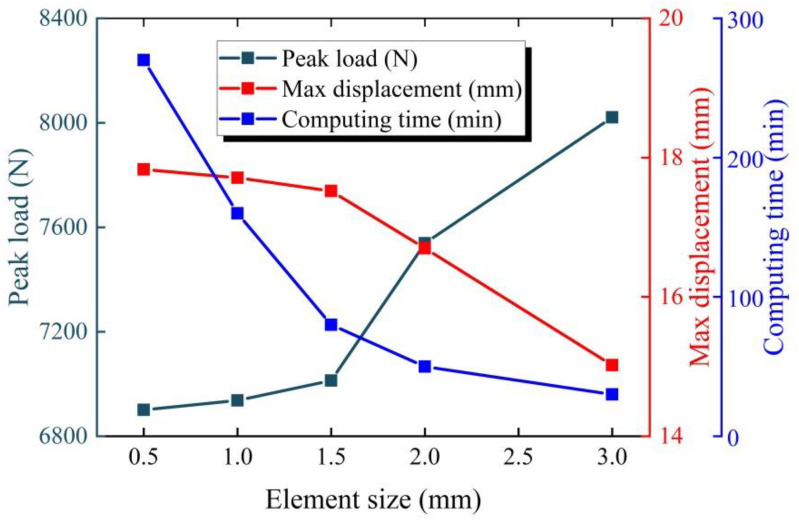
Mesh convergence analysis.

**Figure 4 materials-16-00453-f004:**
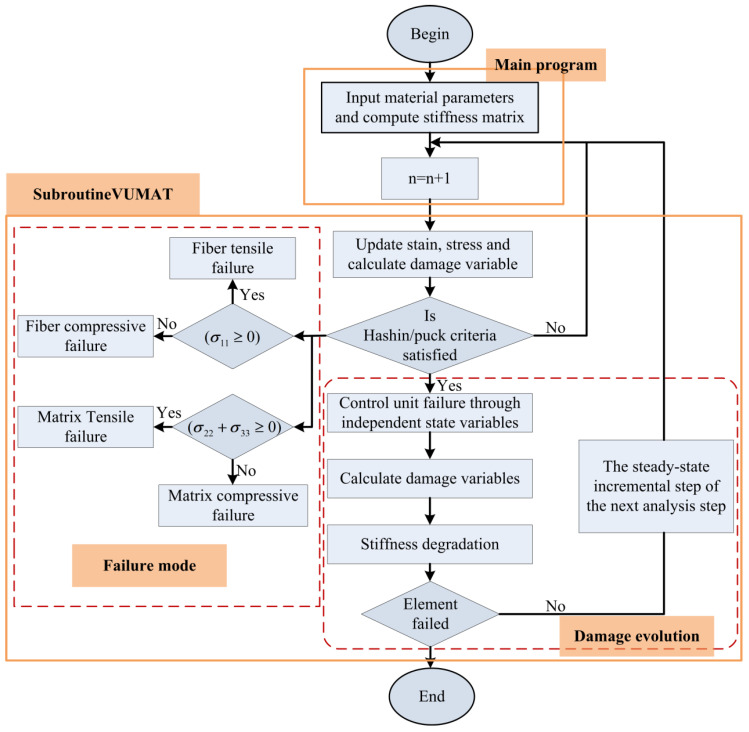
Flowchart for the numerical process of the VUMAT subroutine under ABAQUS.

**Figure 5 materials-16-00453-f005:**
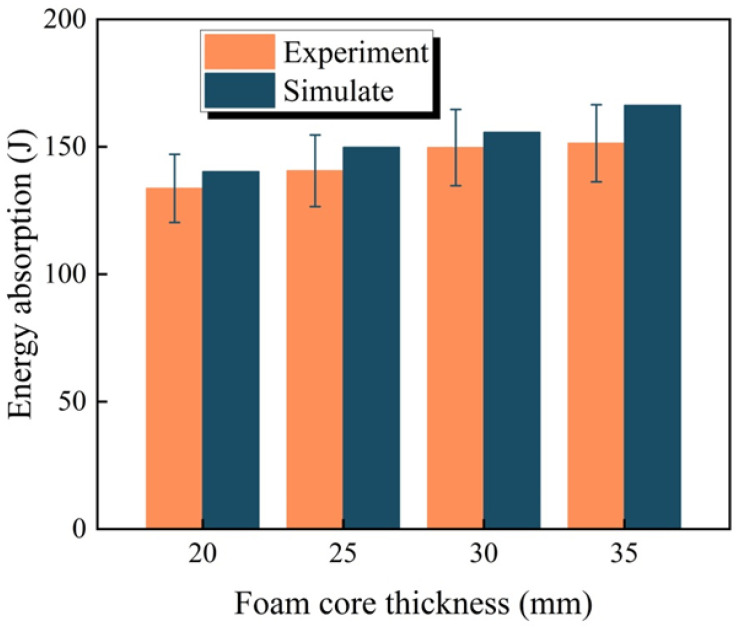
Comparison of CSP’s simulated and experimental impact energy absorption with different foam core thicknesses [25].

**Figure 6 materials-16-00453-f006:**
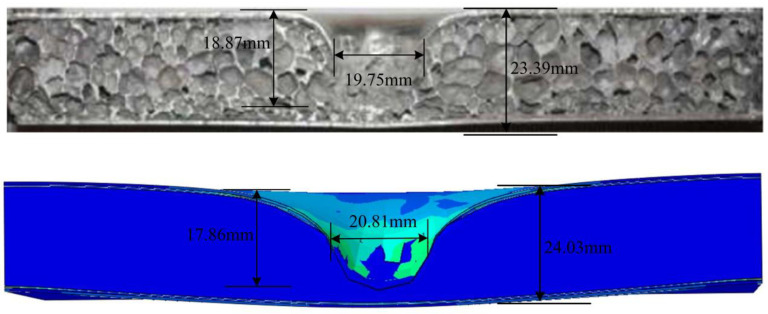
Failure state of the CSP subjected to impact from the experiment and numerical modelling.

**Figure 7 materials-16-00453-f007:**
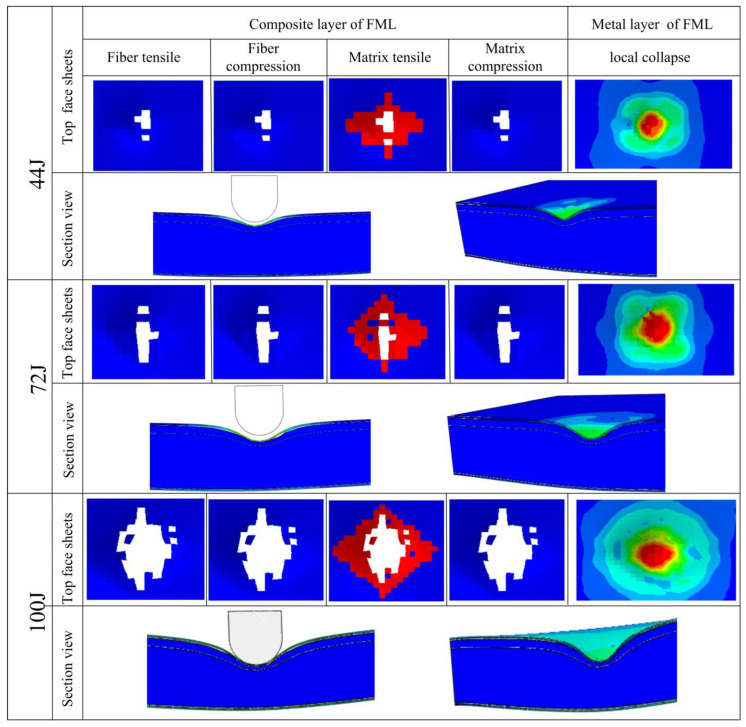
Failure morphologies of the BCSP with different impact energies.

**Figure 8 materials-16-00453-f008:**
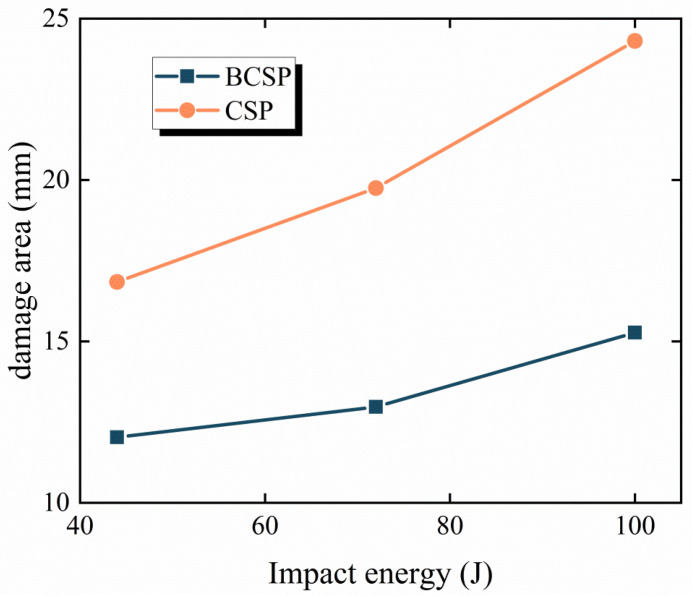
Comparison of the damage area between the BCSP and CSP under different impact energies.

**Figure 9 materials-16-00453-f009:**
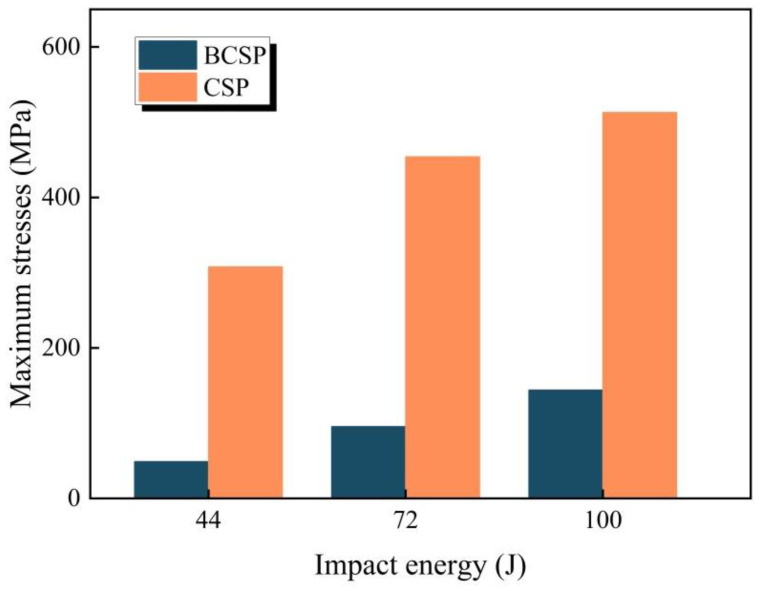
Maximum stresses at the bottom face sheets of the CSP and BCSP under different energies.

**Figure 10 materials-16-00453-f010:**
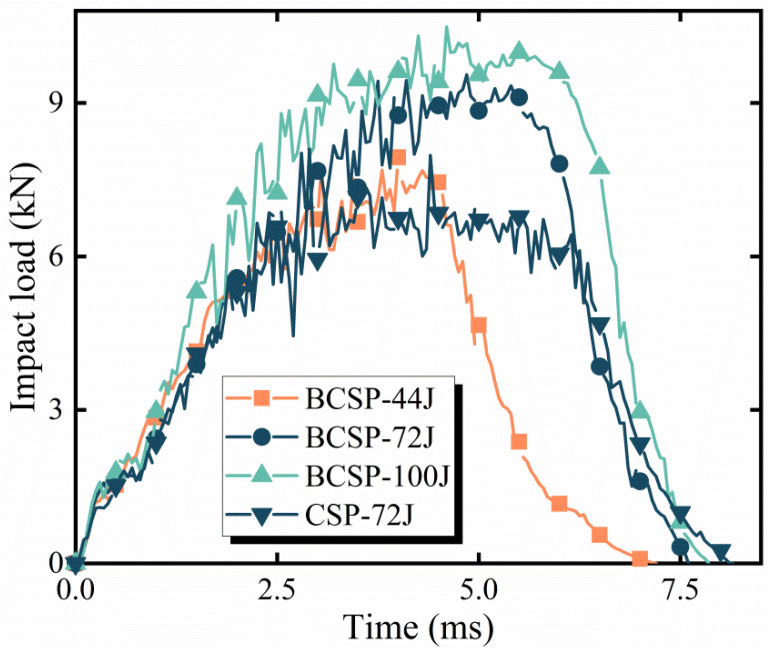
Impact load–time curves of the BCSP.

**Figure 11 materials-16-00453-f011:**
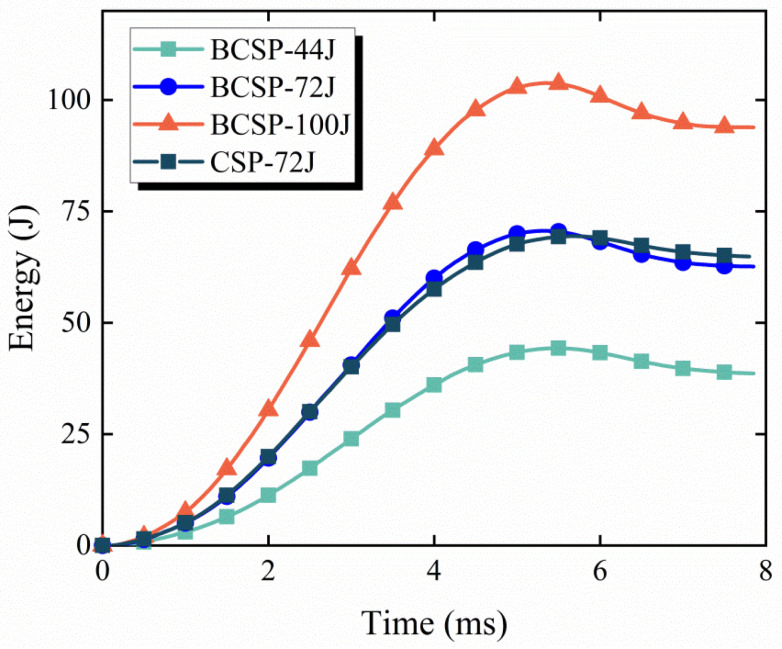
Absorbed energy–time curves of the BCSP.

**Figure 12 materials-16-00453-f012:**
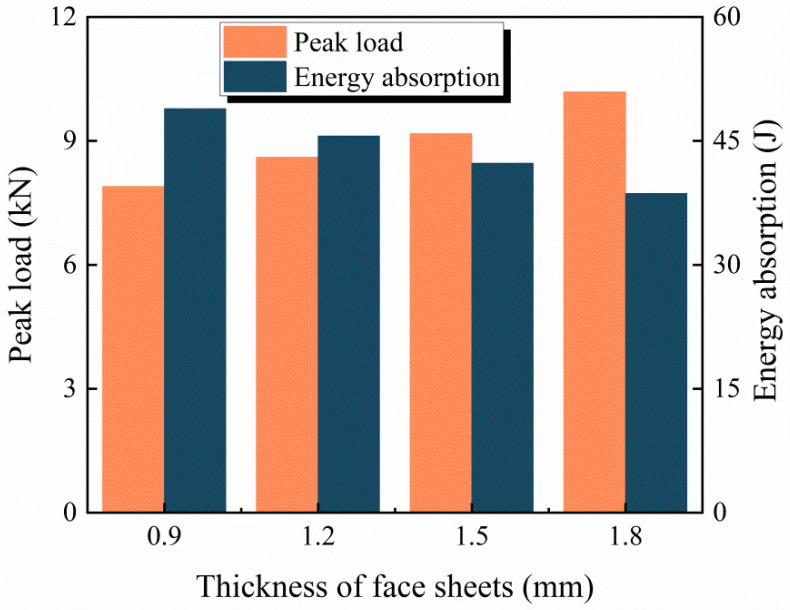
Peak load and impact energy absorption with different face sheet thicknesses.

**Figure 13 materials-16-00453-f013:**
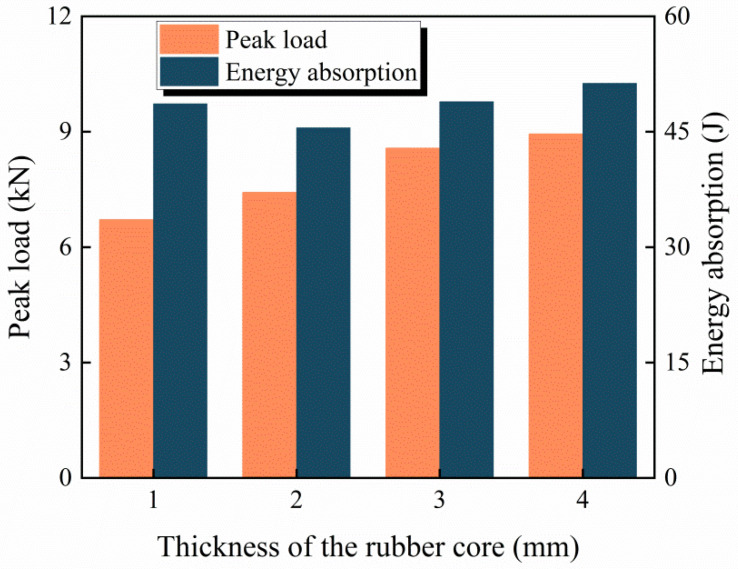
Peak load and impact energy absorption with different rubber core thicknesses.

**Figure 14 materials-16-00453-f014:**
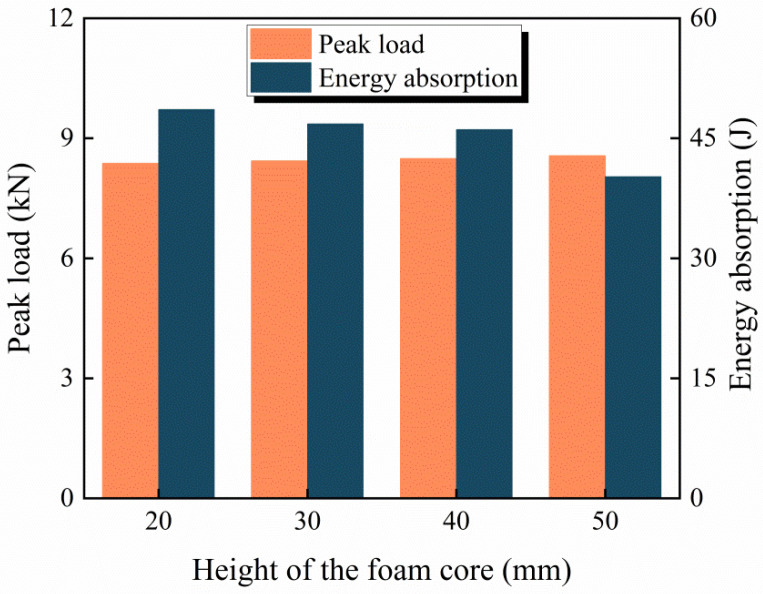
Peak load and impact energy absorption with different foam core heights.

**Table 1 materials-16-00453-t001:** Material parameters of Al 5005 for the FML [40].

Density/kg·m^−3^	Young’s Modulus/MPa	Poisson’s Ratio	Static Yield Limit/MPa
2700	65,000	0.33	171

**Table 2 materials-16-00453-t002:** Material parameters of E glass/epoxy for the FML [41].

Parameter	Numerical Value	Parameter	Numerical Value
Density/kg·m^−3^	1800	Out-of-plane shear modulus/MPa	3530
Longitudinal stiffness/MPa	39,170	Longitudinal tensile strength/MPa	1062
Transverse stiffness/MPa	8390	Transverse tensile strength/MPa	31
Poisson’s ratio	0.38	Transverse compressive strength/MPa	118
In-plane shear modulus/MPa	4140	shear strength/MPa	144

**Table 3 materials-16-00453-t003:** Material parameters of the rubber and Al foam core.

	Al Foam Core [25]	Rubber Core [34]
Density/kg·m^−3^	2700	2000
Elasticity modulus/MPa	37	4
Poisson’s ratio	0	0.49
Tensile stress cut off/MPa	12	-
Damping coefficient	0.1	-
Tensile strength/MPa	-	0.42
Compression yield stress ratio	1.73	-

**Table 4 materials-16-00453-t004:** Summary of the numerical prediction value with various impact energies.

Impact Energy/J	BCSP			CSP
	44 J	72 J	100 J	72 J
Maximum stresses/MPa	48.84	95.45	144.05	453.93
Peak load/kN	7.93	9.52	10.48	7.84
Energy absorption/J	31.2	59.02	89.2	60.62

## Data Availability

All relevant data generated by the authors or analyzed during the study are included within the paper.

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
