# Peer review of "Impact Resistance of a Fiber Metal Laminate Skin Bio-Inspired Composite Sandwich Panel with a Rubber and Foam Dual Core"

_materials, 2023, doi:10.3390/ma16010453_

Round 1
Reviewer 1 Report
The manuscript titled “Impact resistance of fiber metal laminates skin bio-inspired composite sandwich panel with rubber and foam dual-core” brings a mix feeling to me as a reader. This work explores or I can say carefully explores the finite element simulation of bio-inspired (woodpecker) laminated composites over the conventional core-based laminates by assigning different damage models to various layers based on their failure behaviour.
At first the manuscript seems to be interesting, as it is well written with very typographical errors and covers almost every aspect of composite modelling on ABAQUS. However, subsequent reading led to development of multiple queries. Such as, “how reliable was the material properties used in this manuscript (I know it is used from a referred manuscript); can such material with defined interface properties be fabricated and if so, how much deviation we should anticipate in the experimental and analytical values?” If the authors are willing to douse my queries with the response, I suggest the manuscript requires minor revision.
General comments
Excessively long paragraphs. Reduce the length of paragraph since excessively long paragraphs make them boring and diverges the readers.
Page 1, line 9 – “In this paper, a novel bio-inspired….” – Rewrite as “This paper reports the development of a novel…”
Table 2 - Transverse tensile strength – Is it for Z-direction or in Y direction. Since the used stacking sequence (0°/90°) almost gives the similar x and y properties. If it is for Z-direction (thickness) please mention it in the table.
Table 3 – It would be better if the compressive properties of Al foam is provided.
Line 241 - the predicted the damaged area – Change to “the predicted damage area”
Line 268 - mental layer – change to metal layer
Line 290 onwards – Stress transmission – I was wondering why did the authors not add a rubber core before the bottom most FML sheet. I believe this would have further reduced stress in the bottom most ply. Or it was just to make the component economical? Could the authors comment?
Line 340 – What’s BHSP?
Line 383 - the higher the impact resistance it can provide. Please reformat this statement. It’s disturbing the flow of the paragraph.
Line 401-402 - thickness is increases – change to “thickness increases”
Line 402 – what is this BHSB?
Line 410 – In practical engineering…. – this statement requires citation.
Line 450 – “the” is repeated.
Reviewer 2 Report
The low and high-velocity impact on sandwich structures is an important topic in materials and Industrial development. The composite sandwich structures based on the woodpecker’s head design was investigated with finite element method (FEM). Results such as failure morphologies of the sandwich panel, energy absorption and peak load. The work might provide another design for the shielding structure of low-velocity impact. But the work requires essential improvements and some problems should be solved.
1. In the Introduction, Please give a summary of the references and state why this study is needed and how it contributes to the academic research. More recent literature should be referenced.
2. Please give a summary of the references and state why this study is needed and how it contributes to the academic research. a more detailed clarification with the state of the art is requested.
3. The reason for selecting the sizes of specimens dimensions is not supported by any reference.
4. The materials verified with the experiments (ref 25) and model (fig 6) are not exactly computational materials and model (not same core) used in numerical simulation, how to illustrate the results credibly?
5. Add reference of the fig 5.
6. Mesh with Low-order elements such as the four-noded elements, instead of high-order elements, are usually used in the impact analysis. The authors used the C3D8R meshes for the plate discretization. Some explanation on the choice of mesh element should be given. The contact algorithms are critical in simulation of impact phenomena. Details about the contact parameters were also not given.
7. With regard to the details of the finite element meshing, there needs to be considerably more detail presented with full three-dimensional reference than can be discerned from Figure 2, and this needs to be clearly tied to the discussion presented within Section 3. Length scales for the various locations within the model need to be presented along with magnified views so that specifics details can be clearly understood. One particular example at present is the following sentence: “The FML face sheets, Al foam and rubber dual-core of the BCSP were meshed using 3D solid elements C3D8R (the 8-node linear brick, hourglass controlled, reduced integration).”. The details for such need to be clearly established including the mating of such with the model. Beyond all this, the types of elements used throughout need to be clearly described and the reasons for the choices thereof established.
8. More details about the FE model and the analysis should be reported.
9. Page 8 line 232 why unit of energy absorption is 133.7J mm?.
10. The results have been reported in the manuscript but justification is missing.
11. The discussion of the results and drawing of conclusions is rather superficial.
12. The figure 7 is unclear. The quality of figure should be much improved. Make it clearer.
13. The authors reported a few references about the protection sandwich structures under high-velocity impact. More references should be considered the wide diffusion of this topic, including those numerical and experimental work, as an example:
• https://doi.org/10.3390/ma15082899
• https://doi.org/10.1080/15376494.2021.1931991
14. The proposed 2 papers should be inserted in References.
15. Please improve the language used in the manuscript. Some words and sentence are not in form Academic Language.
The reviewer recommends a major revision to improve the overall value of this paper to the engineering community.
Reviewer 3 Report
This paper aims to discuss the impact resistance, under low velocities impact, of a composite panel designed with inspiration on the woodpecker´s head using computational simulations.
The article is well organized and each section is well developed, showing the relevant results with an adequate discussion of them.
Although the paper is well written, some parts of the text must be revisited and rewritten to clarify the information to be passed for the reader. I suggest to better reference the results discussed in paragraph that starts at line 237, as well as figure 6.
I believe that there is a typo in figure 7 at first line last column, perhaps it should be Metal not Mental, as well in lines 268, 318 and 435.
It seems that some information is missing in table 4 and its layout is quite confusing. I recommend to improve it.
Regarding section 4.2, Stress transmission, it is completely unquestionable that the property discussed is far improved, but I think that the results of BCSP and CSP cannot be directly compared, once within the rubber layer the composite layer is thicker than CSP, and there is no easy way to compensate that. Using a foam and/or a metal/epoxy thicker layer will result in new conditions that cannot be directly compared.
In this way, I believe that the paper should be accepted for publishing with minor revisions in MDPI Materials.
Round 2
Reviewer 1 Report
The updated manuscript satisfies the suggested changes and hence could be accepted in the current form.
Reviewer 2 Report
Wenping Zhang et al have thoroughly revised the manuscript ID materials-2080987 entitled « Impact resistance of fiber metal laminates skin bio-inspired composite sandwich panel with rubber and foam dual-core" in accordance with the comments from the reviewers. The paper can be accepted now for publication in its current form in Journal of Materials.